# ACPred: A Computational Tool for the Prediction and Analysis of Anticancer Peptides

**DOI:** 10.3390/molecules24101973

**Published:** 2019-05-22

**Authors:** Nalini Schaduangrat, Chanin Nantasenamat, Virapong Prachayasittikul, Watshara Shoombuatong

**Affiliations:** 1Center of Data Mining and Biomedical Informatics, Faculty of Medical Technology, Mahidol University, Bangkok 10700, Thailand; nalini.schaduangrat@gmail.com (N.S.); chanin.nan@mahidol.edu (C.N.); 2Department of Clinical Microbiology and Applied Technology, Faculty of Medical Technology, Mahidol University, Bangkok 10700, Thailand; virapong.pra@mahidol.ac.th

**Keywords:** anticancer peptide, therapeutic peptides, support vector machine, random forest, machine learning, classification

## Abstract

Anticancer peptides (ACPs) have emerged as a new class of therapeutic agent for cancer treatment due to their lower toxicity as well as greater efficacy, selectivity and specificity when compared to conventional small molecule drugs. However, the experimental identification of ACPs still remains a time-consuming and expensive endeavor. Therefore, it is desirable to develop and improve upon existing computational models for predicting and characterizing ACPs. In this study, we present a bioinformatics tool called the ACPred, which is an interpretable tool for the prediction and characterization of the anticancer activities of peptides. ACPred was developed by utilizing powerful machine learning models (support vector machine and random forest) and various classes of peptide features. It was observed by a jackknife cross-validation test that ACPred can achieve an overall accuracy of 95.61% in identifying ACPs. In addition, analysis revealed the following distinguishing characteristics that ACPs possess: (i) hydrophobic residue enhances the cationic properties of α-helical ACPs resulting in better cell penetration; (ii) the amphipathic nature of the α-helical structure plays a crucial role in its mechanism of cytotoxicity; and (iii) the formation of disulfide bridges on β-sheets is vital for structural maintenance which correlates with its ability to kill cancer cells. Finally, for the convenience of experimental scientists, the ACPred web server was established and made freely available online.

## 1. Introduction

Cancer remains one of the leading causes of death worldwide, accounting for an estimated 18.1 million new cases and resulting in 9.6 million deaths in 2018 [1]. Of these, the most common cancer types causing mortality worldwide include lung, colorectal, stomach, liver and breast cancer, respectively [1,2]. The global burden of cancer has risen dramatically in the last few decades thereby making the prevention of cancer the most significant and challenging public health concern of the 21st century [3]. In spite of the many advanced clinical methods available for the treatment of cancer (e.g., chemotherapy, radiation therapy, hormonal therapy, etc.) their recurrence rate still remains high. In addition, anti-cancer drugs are known to exert their therapeutic effects by killing off normal cells and tissues, which exacerbates the immunodeficiency of patients. Thus, the discovery and development of novel anti-cancer drugs are crucial for decreasing premature death as well as for increasing the survival of many populations. In this regard, peptide-based therapeutics have gained much attention as a new drug class since they are considered to be relatively safe, highly selective, possess good tolerability, lower production cost, easy to modify and synthesize while also exhibiting promising pharmacological profiles [4,5,6,7]. In the last decade, more than 7000 naturally occurring peptides have been discovered and they are found to exhibit a wide range of bioactivities including tumor homing, antihypertensive, antiparasitic, antiviral, antiangiogenic, antibiofilm, antimicrobial and anticancer [8,9,10]. Currently, 60 peptide-based drugs have been FDA-approved [11] with preclinical and clinical studies in the pipeline for another 150 peptides [6].

Anticancer peptides (ACPs) are a subset of antimicrobial peptides (AMPs) that are have been shown to exhibit anticancer activities. These peptides are typically made up of short (5–30 amino acids) cationic residues that mostly adopts the ɑ-helical structure (e.g., BMAP-27, BMAP-28, Cecropin A and B, LL-37, Magainins etc.) or folds into a β-sheet (e.g., Defensins, Lactoferricin and Tachylepsin I) while extended linear forms (e.g., Tritrpticin and Indolicidin) have also been reported [12,13], as shown in Figure 1.

In comparison to normal cells, the cancer cell membrane exhibits different properties (i.e., larger surface area due to the higher number of microvili, high negative charge of the cell membrane, higher membrane fluidity, etc.) [14,15,16,17]. Thus, the inherent cationic property of ACPs is responsible for the electrostatic interactions between ACPs and the anionic cell membrane of cancer cells, and hence allow the selective killing of cancer cells [18]. Previously, many studies have reported that ACPs display several advantages over small molecule drugs. For instance, they exhibit a shorter half-life that consequently leads to decreased probability of resistance, lowered toxicity (not harming normal cells as much), higher specificity, good solubility as well as good tumor penetration capabilities thereby indicating their great potential in cancer therapy [19,20,21,22].

Accurate prediction of ACPs is of great importance for the exploration of their mechanism of action and for the development of therapeutic ACPs. As the experimental identification and development of novel ACPs is a very time-consuming and labor-intensive process, therefore bioinformatics tools are needed for effective analysis of the available big data on existing peptides so as to allow the identification of novel ACPs while also shedding light about their mechanism of action.

Recently, many sequence-based computational methods, for example, AntiCP [23], Hajisharifi et al.’s method [24], ACPP [25], iACP [26], Li and Wang’s method [27], iACP-GAEnsC [28], MLACP [29], SAP [30] and TargetACP [31], have been developed using a wide range of machine learning methods and peptide features as summarized in Table 1. In 2014, Hajisharifi et al. [24] established a benchmark dataset spanning 138 experimentally confirmed ACPs (positive dataset) and 227 non-AMPs (negative dataset). In their method, support vector machine (SVM) with pseudo amino acid composition were used for model building and an accuracy of 92.68% were attained. As noticed in Table 1, iACP [26] was the first computational method based on informative features. In this method, Chen et al. utilized a feature selection technique for obtaining an informative feature set subsequently followed by encoding them for constructing an ACP predictor known as iACP [26] via the use of SVM. This method yielded better prediction accuracy (95.06%) than the two previous methods [23,24]. Thusfar, state-of-the-art ACP predictors includes iACP-GAEnsC [28] and TargetACP [31] in which both afforded high prediction accuracies of 96.45% and 98.78%, respectively. The iACP-GAEnsC [28] method follows the concept of evolutionary intelligent genetic algorithm-based ensemble model for improving the true classification rate. iACP-GAEnsC was developed using various classification algorithms (e.g., random forest, *k*-nearest neighbor, support vector machine, generalized neural network and probabilistic neural network) and three types of peptide features (e.g., pseudo g-gap dipeptide composition, amphiphilic pseudo amino acid composition and reduced amino acid alphabet). More recently, Kabir et al. developed the TargetACP [31] for solving the problem of class imbalance presented in the previously described benchmark dataset [24] by integrating sequential and evolutionary-profile information as input features and using SVM as the classifier. In addition, the authors utilized the synthetic minority oversampling technique (SMOTE) to solve the imbalance phenomenon between minority (ACPs) and majority (non-ACPs) samples.

Although, iACP-GAEnsC and TargetACP yielded encouraging performances with reasonably high prediction accuracies, the overall utility of these two methods is limited in terms of interpretability and practical utility thereby warranting further improvements [32]. One of the major limitations of iACP-GAEnsC and TargetACP was that there was no web server provided for these two methods. Therefore, their utility is limited to researchers with informatics background who can develop in-house prediction models. In addition, the underlying mechanisms of the investigated bioactivity for these two methods affords limited interpretability for experimental scientists. Owing to the heterogeneous computational architecture and design of the study as well as differences in the benchmark dataset used, it is not easy to identify and assess which methods and features provided the greatest contribution to good prediction. Motivated by aforementioned issues presented in previous models [32,33,34,35,36], in order to establish an interpretable ACP predictor we propose a systematic effort for the prediction and analysis of anticancer activities of peptides called the ACPred. Firstly, ACPred was built using various types of peptide features and two powerful machine learning algorithms including random forest (RF) and support vector machine (SVM). Secondly, insights into the ACP mechanism of action were attained via the identified important interpretable features and subsequent in-depth analysis pertaining to the underlying biophysical and biochemical properties of anticancer activities of peptides [33,34,36,37,38]. Thirdly, if-then interpretable rules were extracted using the RF model (IR-ACP). Our prediction results on the benchmark dataset showed that the proposed ACPred achieved an accuracy and MCC of 95.61% and 0.91, respectively, which outperforms most of the existing sequence-based computational methods including Hajisharifi et al.’s method [24], iACP [26], Li and Wang’s method [27], iACP-GAEnsC [28], MLACP [29] and SAP [30]. As previously noted, although iACP-GAEnsC and TargetACP boasts good predictive performances but their interpretability and practical utility prompted the development of ACPred. Thus, ACPred is a user-friendly and publicly accessible web server that allow robust predictions to be made without the need to develop in-house prediction models.

## 2. Materials and Methods

The overall framework of ACPred is illustrated in Figure 2. It can be seen that four major steps are involved in the development of this method as follows: (i) data preparation, (ii) model construction (feature extraction and feature combination), (iii) feature importance analysis and (iv) web server construction. The first step prepares the benchmark dataset, the second step extracts and combines peptide features and the third step analyzes the important features governing ACP prediction via the RF model. Finally, the fourth step involves the construction of a publicly accessible web server using the best model.

### 2.1. Benchmark Dataset

A high-quality benchmark dataset can ensure the reliability of the proposed method. Therefore, the investigated dataset was retrieved from the work of Hajisharifi et al. [24], which contains 192 ACPs and 215 non-ACP. As to guarantee the quality of the benchmark dataset, the following steps were considered. Firstly, in order to avoid a dataset containing redundant peptides, the CD-HIT program was used to eliminate ACPs and non-ACPs with >90% identity. Secondly, ACPs and non-ACP containing special characters, such as X and U, were removed. After such screening procedures, a set of 138 ACPs and 205 non-ACPs was attained for prediction and analysis of ACPs.

### 2.2. Feature Representation

In order to build an effective prediction model, it is necessary to represent each peptide as numerical vectors that can encompass the perspective of its biophysical and biochemical properties. For peptide or protein sequences, the most classical and interpretable features are amino acid composition (AAC), dipeptide composition (DPC) and physicochemical property (PCP).

According to the classical definition, AAC and DPC are the proportions of each amino acid and dipeptide in a peptide sequence **P** that are expressed as fixed lengths of 20 and 400, respectively. Thus, in terms of AAC and DPC features, a peptide **P** can be expressed by vectors with 20D and 400D (dimension) spaces, respectively, as formulated by the following two equations:(1)$$P={\left[aa_{1},aa_{2},\ldots ,\;aa_{20}\right]}^{T}$$
(2)$$P={\left[dp_{1},dp_{2},\ldots ,\;dp_{400}\right]}^{T}$$
where *T* is the transpose operator, while *aa_1_*, *aa_2_*, …, *aa_20_* and *dp_1_*, *dp_2_*, …, *dp_400_* are occurrence frequencies of the 20 and 400 native amino acids and dipeptides, respectively, in a peptide sequence **P**. PCP is one of the most intuitive features associated with biophysical and biochemical reactions. In fact, there are 544 PCPs of amino acids extracted from the amino acid index database (AAindex) [39], which is a collection of published literature as well as different biochemical and biophysical properties of amino acids. Each physicochemical property consists of a set of 20 numerical values for amino acids. After removing 13 PCPs with not applicable (NA) as their amino acid indices, a total of 531 PCPs were further used in this study.

As mentioned in the study of [40,41,42], the sequence order information of AAC and DPC would be lost. To deal with such a dilemma, the pseudo amino acid composition (PseAAC) and amphiphilic pseudo amino acid composition (Am-PseAAC) approaches were proposed. According to Chou’s PseAAC [41], the general form of PseAAC for a peptide **P** is formulated by: where the subscript Ω is an integer to reflect the feature’s dimension. The value of *Ω* and the component of $\Psi_{u}$, where u=1,2,…,Ω depends on the protein or peptide sequences. In this study, the parameters of PseAAC, i.e., the discrete correlation factor λ and weight of the sequence information ϖ, were set by default in the protr R package [43]. The dimension of PseAAC feature is 20 + λ×ϖ. Since the hydrophobic and hydrophilic properties of proteins play an important role in its folding and interaction of proteins, Am-PseAAC was introduced by Chou [40]. The dimension of Am-PseAAC feature is 20 + 2λ whereby, the first 20 components are the 20 basic AAC ($p_{1},p_{2},\ldots ,\;p_{20}$) and the next 2λ ones are a set of correlation factors that reveal the physicochemical properties such as hydrophobicity and hydrophilicity along a protein or peptide sequence as formulated by:(3)$$P={\left[\Psi_{1},\Psi_{2},\ldots ,\Psi_{u},\ldots ,\;\Psi_{\Omega }\right]}^{T}$$
where the subscript *Ω* is an integer to reflect the feature’s dimension. The value of *Ω* and the component of $\Psi_{u}$, where u=1,2,…,Ω depend on the protein or peptide sequences. In this study, the parameters of PseAAC, i.e., the discrete correlation factor λ and weight of the sequence information ϖ, were set by defaults using the protr R package [43]. The dimension of PseAAC feature is 20+λ×ϖ. Since hydrophobic and hydrophilic properties play an important role in the folding and interaction of proteins, Am-PseAAC was introduced by Chou [40] for representing such features. The dimension of Am-PseAAC feature is 20+2λ whereby the first 20 components are the 20 basic AAC ($p_{1},p_{2},\ldots ,\;p_{20}$) while the next 2λ
ones are a set of correlation factors that pertains to physicochemical properties namely hydrophobicity and hydrophilicity along a protein or peptide sequence as formulated by:(4)$$P={\left[p_{1},p_{2},\ldots ,\;p_{20},p_{20+\lambda },p_{20+\lambda +1},\ldots p_{20+2\lambda }\right]}^{T}$$

In this study, the five aforementioned features of peptide sequences were calculated using the protr package in the R software [43].

### 2.3. Dataset Modelability

The prediction performances of QSAR models are directly influenced by the dataset and the types of feature used. Recently, the concept of dataset modelability has been proposed by Golbraikh et al. [32,35,44,45] for evaluating the feasibility of developing a robust and predictive model for a given dataset of interest. In this study, the MODI score for each type of peptide features and their combinations were calculated using an in-house developed R code. Prediction models are recommended for further analysis if the MODI score is greater than 0.65; otherwise, prediction models are not recommended. Further details pertaining to the calculation of the MODI score can be found elsewhere [44].

### 2.4. Classifier Selection

In this study, we employed two successful machine learning algorithms namely random forest (RF) and support vector machine (SVM). Previously, these approaches have been successfully used in the prediction of various functions and properties of peptides and proteins [33,36,38,46,47,48,49,50,51] as well as other biological or chemical entities [35,45,52,53,54,55]. Herein, the basic concepts and associated parameter optimizations for the two classifiers are briefly described hereafter.

RF models were constructed according to the described original RF algorithm [56]. These models were developed by growing many weak classification and regression tree (CART) classifiers where each classifier is generated using a random vector sampled independently from the input vector as to enhance the prediction performance of CART [56,57]. In the RF method, the out-of-bag (OOB) approach is used for evaluating the feature importance as follows: (1) two-thirds of the training data was utilized to construct the predictive classifier while the remaining was used for evaluating the performance of such classifier and (2) the feature importance of each feature can be evaluated by measuring the decrease in the prediction performance. It should be noted that the performance evaluation of the model can be either accuracy or Gini index. Herein, the RF classifier was established using the *randomForest* package in the R software [58]. To enhance the performance of the RF model, two parameters namely, *ntree* (i.e., the number of trees used for constructing the RF classifier) and *mtry* (i.e., the number of random candidate features) were determined using the *caret* package in the R software [59] via a 5-fold cross-validation (5-fold CV) approach. The search space of *ntree* and *mtry* were [100, 500] and [1, 10] using steps of 100 and 1, respectively.

SVM is a supervised learning model based on the principles of structure risk minimization and kernel method as proposed by Vapnik [60,61,62]. SVM model can deal with the problem of over-fitting arising with the use of a small training dataset by mapping the input samples to a higher dimension space and then searching for the maximum-margin hyperplane for constructing the classifier [63,64]. Previously, SVM models were used in many applications because of their predictive performance capabilities when applied to classification, prediction, and regression problems. In the optimization process, the regularization parameter C∈{0.25, 0.50, 1, 2, 4} was determined with a 5-fold CV approach using the *caret* package in the R software [59].

### 2.5. Identification of Important Features

The analysis and identification of feature importance can provide a better understanding on the underlying biophysical and biochemical properties governing anticancer activities of peptides. Herein, the efficient and effective built-in feature importance estimator of the RF method was used to reveal and characterize differences between ACPs and non-ACPs. As mentioned above, the RF method provides two measures for ranking feature importance, i.e., the mean decrease of the Gini index (MDGI) and the mean decrease of prediction accuracy. Since Calle and Urrea [65] demonstrated that the MDGI provided more robust results when compared with the mean decrease of prediction accuracy, we utilized the MDGI to rank the importance of interpretable features that included AAC, DPC and PCP. Until now, these three features have been used to characterize many peptides and proteins such as for predicting HIV-1 CRF01-AE co-receptor usage [46], predicting protein crystallization [51,66], predicting the oligomeric states of fluorescent proteins [38], predicting the bioactivity of host defense peptides [47], prediction of human leukocyte antigen gene [37,49], predicting antifreeze proteins [33], predicting the hemolytic activity of peptides [34] and predicting the antihypertensive activity of peptides [36].

The MDGI is an impurity measure that corresponds to the ability of each feature in discriminating the sample classes. The Gini index can be defined as follows:(5)$$1-\overset{2}{\underset{c=1}{\sum }}p^{2}(c|t)$$
where denotes the estimated class probability for node t in a tree classifier and c is the class label (i.e., either ACPs or non-ACPs). Features with the largest MDGI value is considered to be an important feature as it significantly contributes to the prediction performance.

### 2.6. Rule Extraction

This work presented an interpretable rule extraction of ACPs (IR-ACP) based on the RF method cooperating with AAC features for determining how the features work in combination within the learning process. A set of rules from an individual tree is derived from the root to the leaves. In this study, only 100 decision trees were used to extract if-then interpretable rules for explaining the prediction results by means of the *RRF*, *inTrees* and *xtable* packages in the R software [56,57,67]. More details of the rule extraction process can be found in previous related works [67,68].

### 2.7. Performance Evaluation

Three testing methods consisting of (i) sub-sampling test (2-, 5- or 10-fold cross-validation), (ii) jackknife test or also known as leave-one-out cross-validation (LOO-CV), and (iii) independent (or external) testing dataset are often used to evaluate the prediction performance. The sub-sampling test is one of the popular cross-validation methods for assess the predictive capability of a model. As elucidated in [41,69] and demonstrated by Equation (50) of [70], among the three cross-validation methods, the jackknife test is considered as one of the most rigorous and objective methods for cross-validation in statistics and can provide a unique result for a given benchmark dataset. Thus, in this study, the 5-fold cross-validation (5-fold CV) and jackknife test were used to evaluate the prediction performance of our models. The former set (5-fold CV) makes use of data from the training set where the data set is separated into five subsets. Practically, one subset from a total of five subsets is left out as the testing set while the remaining are used for training the model. This process is repeated iteratively until all data samples have had the chance to be left out as the testing set. During the jackknifing process, each sample in the dataset (N samples) is left out as an independent sample (one sample) and the remaining sequences (N-1 samples) are used for training the model. This process is repeated iteratively until all samples have had the chance to be left out as the independent sample.

In order to evaluate the prediction ability of the model, the following sets of four metrics are used:(6)$$Ac=\frac{TP+TN}{\left(TP+TN+FP+FN\right)}$$
(7)$$Sn=\frac{TP}{\left(TP+FN\right)}$$
(8)$$Sp=\frac{TN}{\left(TN+FP\right)}$$
(9)$$MCC=\frac{TP\times TN-FP\times FN}{\sqrt{\left(TP+FP\right)\left(TP+FN\right)\left(TN+FP\right)\left(TN+FN\right)}}$$
where *Ac*, *Sn*, *Sp* and *MCC* represent accuracy, sensitivity, specificity and Matthews coefficient correlation, respectively. *TP*, *TN*, *FP*, and *FN* represent the instances of true positive, true negative, false positive and false negative, respectively. Moreover, in order to evaluate the prediction performance of models using threshold-independent parameters, receiver operating characteristic (*ROC*) curves were plotted using the pROC package in the R software [71]. The area under the ROC curve (*auAUC*) was used to measure the prediction performance, where *AUC* values of 0.5 and 1 are indicative of perfect and random models, respectively.

### 2.8. Reproducible Research

To ensure the reproducibility of models proposed herein, all R codes and the benchmark dataset used in the construction of the predictive models, graphical figures and the ACPred web server are available on GitHub at https://github.com/Shoombuatong2527/acpred and https://github.com/chaninlab/acpred-webserver.

## 3. Results and Discussion

### 3.1. Prediction Performance

In this study, effort is directed toward the determination of feature types that are beneficial for the prediction of APCs. This was performed by comparing the performance of different feature types as constructed using SVM and RF by means of 5-fold CV and jackknife test. The five basic features consisting of AAC, DPC, PCP, PseAAC and Am-PseAAC were considered as well as their combinations: AAC + PseACC, AAC + Am-PseACC, PseACC + Am-PseACC and AAC + PseACC + Am-PseACC, which were also utilized to investigate the complementarity of the five aforementioned features.

Prior to the development of the predictive model, the assessment of modelability of the benchmark dataset was performed by computing the MODI index. This index helps modelers to estimate the feasibility of obtaining robust and reliable predictive models. For the binary classification problem, if the value of MODI index is greater than 0.65, the feature is considered to be reliable for classification modelling; otherwise, the feature is not recommended for classification modelling.

As shown in Table A1, all feature types and their combinations from the benchmark dataset met these criteria. Thus, it could be stated that our proposed features are reliable and efficient for constructing ACP predictor.

Table 2 and Table 3 shows the performance comparison amongst different feature types with RF and SVM on the benchmark dataset via 5-fold CV and jackknife test, respectively. In addition, Figure 3 shows the receiver operating characteristic (*ROC*) curve of the five considered feature sets obtained from RF (top and bottom left) and SVM (top and bottom right) as evaluated by 5-fold CV (top left and right) and jackknife test (bottom left and right). As seen in Table 2 and Table 3, SVM model with the Am-PseAAC feature afforded the highest accuracy of 95.03% as evaluated by the jackknife test. Meanwhile, RF model with PseACC feature and SVM model with AAC feature performed well with the second and third the highest accuracy of 93.28% and 92.98%, respectively. These results showed that ACC, PseACC and Am-PseAAC were effective for ACP prediction. Since using a combination of various features might yield better prediction performance, four combinations of the three effective features were also considered. The highest accuracy and *MCC* of 95.61% and 0.91, respectively, was achieved by using SVM model cooperating with the combination feature of AAC and Am-PseAAC, while using the combination of the three effective features, i.e., ACC, PseACC and Am-PseAAC, performed slightly worse than the combination feature of AAC and Am-PseAAC with an accuracy and *MCC* of 94.74% and 0.89, respectively. In addition, our prediction results were well consistent with previous studies [27,28,29] and related studies [33,34,36,72].

Furthermore, observations pertaining to the performance comparisons from Table 2 and Table 3 and Figure 3, it can be briefly summarized as follows. Table 2 and Table 3 shows that both RF and SVM models afforded improved prediction performances when using the Am-PseACC feature while the best prediction performance on the benchmark dataset via both 5-fold CV and jackknife test were achieved by the SVM model trained using the combination of AAC and Am-PseAAC features.

For the convenience of subsequent descriptions, we will refer to this method as ACPred as it represents the best model that will be used for further comparisons with other tools.

### 3.2. Comparison with Other Methods

To indicate the effectiveness of the proposed method, we compared the performance of our selected model (named ACPred) with other popular ACP predictors. Since Hajisharifi et al. [24] were the first group to establish the benchmark dataset, and iACP [26], iACP-GAEnsC [28] and TargetACP [31] were state-of-the-art ACP predictors that provided prediction results as assessed by the jackknife test, therefore these predictors were used for performance comparisons. Table 4 lists the performance comparisons of ACPred with the three ACP predictors. As noticed from Table 4, prediction results were obtained from two different experimental designs: (i) the prediction model was performed on the benchmark dataset as derived from the work of [24] as well as those of Hajisharifi et al.’s method, iACP, iACP-GAEnsC, TargetACP and ACPred, (ii) the prediction model was built using the balanced dataset including TargetACP and ACPred. The reported results from aforementioned methods as summarized in Table 4 was obtained directly from the work of iACP-GAEnsC [28].

As for performance comparisons on the benchmark dataset, it was found that ACPred was comparable with that of iACP-GAEnsC and TargetACP as indicated by four statistical parameters. Moreover, ACPred yielded a greater prediction performance than the method proposed by Hajisharifi et al. and iACP. To the best of our knowledge, iACP-GAEnsC is the only method that utilizes the synthetic minority oversampling technique (SMOTE) [73] technique for constructing a balanced dataset from the original benchmark dataset [24]. From the perspectives of machine learning, it is not fair to directly compare prediction results between TargetACP and ACPred because TargetACP was trained and tested on the balanced dataset consisting of 205 ACPs and 205 non-ACPs. In order to perform a fair comparison, the herein proposed ACPred method was applied on the balanced dataset (called ACPred-modified), which was generated by means of the SMOTE technique. Herein, the oversampling method was used to add synthetic samples for the minority class (i.e., ACPs) presented in the benchmark dataset by setting the parameters of the number of nearest neighbours (*k*), the number of extra cases (oversampling) to add to the minority class (perc.over) and the number of cases to reduce (undersampling) from the majority class (perc.under) were set to 9, 50 and 300, respectively. As seen in Table 4, ACPred-modified was found to outperform Hajisharifi et al.’s method, iACP and iACP-GAEnsC with an improvement of 2%–4% and 3%–7% on *Ac* and *MCC*, respectively. However, it was observed that ACPred-modified performed slightly worse than TargetACP by approximately 2%–3% where ACPred-modified and TargetACP provided 97.56% *Ac*/0.95 *MCC* and 98.78% *Ac*/0.97 *MCC*, respectively. Nevertheless, TargetACP was constructed with a sophisticated design thus, it is not easy to identify and assess which features offer the most contribution to the prediction improvement. On the other hand, ACPred was designed in a systematic manner for prediction and characterization of anticancer activities of peptides. In addition, a user-friendly web server ACPred was developed to facilitate high-throughput prediction of ACP. Therefore, the proposed ACPred model could become a practical tool for predicting and interpreting the anticancer activity of peptides, or at least as a complementary tool to existing methods in the field.

Thinking to the future, some possible improvements that could enhance the prediction performance for predicting anticancer activity of peptides are described hereafter. Firstly, it is worthy to explore the separation of peptides according to their sequence lengths (e.g., peptides having 4, 5 or 6 amino acids, peptides having 7–12 amino acids, and peptides having more than 12 amino acids) followed by the development of separate predictive models for each of the five sequence range [36,74]. Secondly, extract 5, 10 and 15 amino acids from the N or C terminus for the development of predictive QSAR models, which has successfully been demonstrated by the group of P.S. Raghava [23] Thirdly, explore the modelability of the dataset using the approach of Golbraikh et al. [44] such that robust models could be developed. Fourthly, the currently employed non-anticancer peptides [24] are not based on true experimentally-determined inactive peptides but are random peptides obtained from the UniProt database therefore, if such true inactive peptides could be determined they may aid in the development of robust models.

Aside from the aforementioned methodological improvements that have been witnessed from the literature in the development of anticancer peptide classifiers, however, the underlying origin of anticancer activity of investigated peptides as rationalized by peptide features is an area that deserves more attention. Current predictive models used for the prediction of anticancer activity of peptides are primarily based on sequence order-independent descriptors that does not consider the order of amino acids in a peptide sequence (e.g., AAC and DPC), which hampers the identification of motifs from peptides. Such limitation has been addressed by the group of KC Chou [41,75,76] in which they developed PseAAC and Am-PseAAC descriptors that take into account sequence order information based on physicochemical properties. However, the interpretability of these descriptors is limited and thus, existing approaches for rationalizing biological interpretation from these predictive models are resorted back to the sequence order-independent descriptors such as AAC and DPC. Analysis of the literature revealed that interpretable descriptors such as the *z*-scale descriptors [77,78] have been successful for predicting robust and interpretable models, however the flaw of this approach is that the peptides used in the development of the model should be of equal length so as to conduct sequence alignment. Methodological tweaks addressing such limitation have been made via the use of autocorrelation such that sequence alignment is not needed and would allow the models to be built on peptides of varied lengths. In spite of this, this method is not applicable in this study field in which the prepared dataset removes peptides having high similarity of greater than 90% therefore, rendering the removal of a large number of peptides that is normally needed when using *z*-scale descriptors. As such, there is ample room for the discovery of new and interpretable descriptors for the development of predictive models of peptides, which is an important area for further driving the field of anticancer peptide prediction.

### 3.3. Biological Space

The identification of feature importance from AAC and DPC descriptors can provide a better understanding on the biochemical and biophysical properties of anticancer activities of peptides. Previously, AAC and DPC features have been analyzed as to further gain insights on how to characterize therapeutic peptides [29,34,36,47] and protein functions [33,51,79]. In this study, the value of MDGI was adopted to rank and estimate the importance of each AAC and DPC feature. Such information is derived from the analysis of the benchmark dataset that consists of 138 ACPs and 205 non-ACPs. Table 5 lists the percentage values of the 20 amino acids for both ACPs and non-ACPs along with amino acid composition difference between the two classes as well as their MDGI values. In addition, a heatmap showing the feature importance for DPC features is shown in Figure 4. From Table 5, it can be observed that the ten informative amino acids with the highest MDGI values are Lys, Cys, Gln, Arg, Ile, Leu, His, Pro, Glu and Gly (29.54, 19.71, 15.89, 11.42, 10.13, 7.84, 7.80, 7.37, 6.88 and 6.56, respectively). Meanwhile, Figure 4 shows that the 20 top-ranked dipeptides according to their MDGI value are KK, KI, CY, IK, AK, KV, IP, LK, TC, SC, CG, LF, FK, GL, FL, LR, PP, LG, IG and AQ. Interestingly, among the 5 top-ranked informative amino acids, 3 of these are polar (i.e., Lys, Gln and Arg), while 2 are non-polar (i.e., Cys and Ile) and hydrophobic residues. In addition, Lys and Arg are also positively-charged residues, which may support the anti-cancer cell penetrating properties of peptides (i.e., that is required for targeting tumors with specificity and low toxicity) [80]. Although the percentage compositions differentiating ACPs from non-ACPs as presented in Table 5 highlight that ACPs a have higher percentage of polar residues and lower percentage of basic residues as compared to non-ACPs, cell surface binding and internalization are critical for specific targeting of cells with anti-cancer activities. According to the membrane of cancer cells presenting anionic molecules such as phosphatidyl serine (PS), heparin sulfate and O-glycosylated mucins, the ACPs that contain positive charge are critical for endocytosis and selectively killing cancer cells. While hydrogen bonds that appear on ACPs were designed for improving the solubilization of hydrophobic molecules on ACPs. However, positively charged amino acids alone are not enough to completely neutralize cancer cells but rather aid in rapid internalization of cancer cells for selective killing with low toxicity to normal cells. Moreover, hydrogen bonded interactions to encapsulate anti-cancer drugs has been utilized for increasing the oral efficiency and drug-controlled release. Therefore, only hydrogen bond on ACPs are not adequate to resist cancer cells. Thus, the combination between crucial amino acids and physicochemical properties could provide a synergistic effect that enhances the efficiency of ACPs [81]. As such, results from the analysis of the top ranked informative AAC are discussed below.

To roughly analyze the characteristics of ACPs, a few studies have shed light on important amino acids and dipeptides commonly found in ACPs by using simple composition analysis approaches without the use of experimental methods. For instance, Tyagi et al. [23] reported results of residue preference at 10 N-terminus and 10 C-terminus by using the sequence logos. Their analysis revealed that Leu and Lys were typically found at the N-terminus while Cys, Leu and Lys were typically found at the C-terminus. Chen et al. [26] also confirmed that amino acids including Lys, Ile, Cys, Glu, and Gly were abundant in ACPs as compared to non-ACPs. Recently, Manavalan et al. [29] revealed that the three top-ranked informative dipeptides consisted of KK, AK and KL. As can be noticed in Table 5 and Figure 4, our analysis results derived from the MDGI values of AAC and DPC were quite consistent with the three aforementioned studies.

As mentioned above, amongst the five top-ranked informative amino acids, Lys had the highest MDGI value of 29.54. Such feature with the highest MDGI is considered to be the most important factor governing the anticancer activity. As a basic residue, Lys is highly conserved in the composition of therapeutic peptides as it enhances the formation of electrostatic interactions between the peptide and the plasma membrane owing to its cationicity [82]. The role of cationicity in ACPs have been previously investigated by the study of Gopal et al. [83]. The authors modified a synthetic AMP known as HPA3NT3 (FKRLKKLFKKIWNWK), which was derived from Helicobacter pylori by substituting Arg and Asn with Lys at positions 3 and 13, respectively. Gopal et al. reported that the Lys-modified peptide increased the peptide selectivity toward negatively-charged membrane surface, which could allow for endocytosis and specificity of the APC in killing cancer cells. In addition, Wang et al. [84] has shown that the cell-penetrating ability is critical for specifically targeting cationic cancer cells. The authors observed that a Lys-rich peptide, L-K6 (IKKILSKIKKLLK-NH2) could internalize into MCF-7 cells without significant cell surface disruption. Interestingly, the authors also found that the internalized L-K6 could induce MCF-7 cell death without significant cytoskeleton disruption and mitochondrial impairment. Moreover, these findings indicated that the negatively-charged phosphatidylserine (PS) reported to be abundantly exposed on cancer cell surfaces [85,86] might contribute to the preferential binding of L-K6 to MCF-7 cancer cells. Furthermore, defensins are a large subfamily of natural cationic human AMP showing anticancer properties. Their roles in the immunomodulation of the innate and adaptive immune system have extensively been studied in previous works [87,88]. Defensins are categorized as either α-defensins or β-defensins, depending on their sequence homology and Cys connectivity (Figure 1), they are found to be rich in cationic residues (e.g., Lys and Arg) that enhances their immunomodulatory effects [89]. Papo et al. [90] constructed a short 15 amino acid peptide consisting of d- or l- amino acids (d-K_6_L_9_; LKLLKKLLKKLLKLL-NH2) and observed that d-K_6_L_9_ specifically targets and lyses cancer cells as well as inhibiting the growth of primary and metastatic tumors in in vivo models. The ability of d-K_6_L_9_ in selectively targeting cancer cells can be attributed to its cationicity whereby the peptide can bind to highly enriched surface-exposed PS in cancer cells as compared to non-cancer cells [91]. Similarly, Arg, which is a positively-charged residue that differs from Lys by its side chain, was shown to be the fourth informative amino acid form our analysis thus, reiterating the importance of the cationic property of APCs.

The second most important amino acid was Cys that afforded an MDGI value of 19.71. Cys is a highly reactive amino acid in which it contains a thiol group that is oxidized to form a dimer thus creating a disulfide bridge between two spatially adjacent Cys residues. The significance of disulfide bridges are seen in their strong, hydrophobic nature which are extremely important for the overall structural fold of the peptide, while also increasing the peptide stability against proteolytic degradation [92]. Almost a decade ago, GO-201 ([R]_9_CQCRRKNYGQLDIFP) achieved in facilitating the inhibition of mucin-1 cytoplasmic domain (MUC1-CD) [93]. Thus far, many studies have established the function of the GO-201 peptide whereby its anticancer activity was demonstrated both in vitro and in vivo against human breast cancer, prostate cancer, chronic myelogenous leukemia and pancreatic cancer [93,94,95,96]. From these aforementioned studies, the researchers were able to determine key motifs responsible for the anticancer activity of GO-201 as CxC, thus reiterating the crucial role of Cys disulfide bridges [97]. Furthermore, Tyuryaeva et al. [98] explored the molecular mechanism of Cys-induced cytotoxicity and the resulting apoptosis of cancer cells by designing and testing peptide sequences containing the key CxC ([R]_9_KCGCFF; named DIL) or CxxC ([R]_9_FF**C**PH**C**YQ; named DOL) motifs. The results obtained from this study suggests that the disulfide oxidoreductase is responsible for the effective cytotoxtic action of Cys. Therefore, it can be inferred that peptides containing the CxC or CxxC motifs form disulfide bridges that pilot a partial loss of protein function and the rapid onset of apoptosis. It can therefore be anticipated that such peptide sequences could help to create a potentially valuable new class of ACPs. Similarly, Schroeder et al. [99] ascertained a novel peptide GKAKCCK having pronounced antimicrobial activity. The authors discovered that the efficiency of this peptide was critically dependent on the two continuous cysteines and that upon mutating these residues to Ala or Ser, its activity was completely abrogated. Hence, given the well-established connection between antimicrobial and anticancer peptides [13,100], it is likely that the GKAKCCK peptide is able to induce antitumorigenic effects.

### 3.4. Mechanistic Interpretation of Informative PCP

The physicochemical properties of amino acids play an essential role as effective features for identifying and characterizing protein or peptide functions from their primary sequences [33,34,36,51,79]. It is well known that PCPs [39], such as molecular volume, exposure or accessible surface, polarity (hydrophobicity/hydrophilicity), charge/pK, hydrogen-bonding potential and so forth are correlated with the structure and function of the amino acid sequence [101]. Herein, we have obtained the ten top-ranked informative PCPs corresponding to their MDGI values as shown in Table 6.

The most important PCP is seen to be ‘hydrophobicity’ with an MDGI value of 1.51, which corresponds to the AAindex ARGP820101. In addition, from the analysis, 4 of the 10 top-ranked informative PCPs are related to α-helix properties of peptides consisting of ARGP820102, BEGF750101, ISOY800106 and CHOP780201 with corresponding MDGI values of 1.40, 0.92, 0.64 and 0.61, respectively. Furthermore, 3 of 10 top-ranked informative PCPs including the AAindices of BEGF750102, BEGF750103 and CHOP780101 are related to β-sheet properties with MDGI values of 1.00, 0.87 and 0.83, respectively. Consequently, it could be inferred that the hydrophobicity of a peptide along with its structural orientation (i.e., α-helix or β-sheet) are significant in determining the anticancer properties of peptides. The important PCPs of ACPs are analyzed and discussed below.

#### 3.4.1. Hydrophobic Residues on α-Helical Structure Enhances Cell Penetrating Properties of Anticancer Peptides

It is commonly known that the hydrophobicity property is critical for the 3D structure of a protein. Many descriptors have been developed for measuring the hydrophobicity of proteins, depending on the presence of hydrophobic residues at specific locations in the peptide sequence. These indices are usually based on the 3D crystal structure of proteins coupled with the physicochemical properties of their side chains [102,103]. For instance, Argos et al. [104] utilized the hydrophobic index as measured by Nozaki and Tanford [105] for the 20 canonical amino acids for determining the role of hydrophobicity on the structure of amino acids. As mentioned above, the property of ARGP820101 with an MDGI value of 1.51 pertained to the hydrophobicity property. This result suggests that the presence of hydrophobic amino acid residues on the α-helix might be important for governing the anticancer activity of peptides. Moreover, the property of ARGP820103 (the fourth most important PCPs) is also associated with hydrophobicity property. Many previous studies have described the need for symbiosis between hydrophobicity and cationicity of peptides in order to enhance its cell penetrating property and its ability to reach target cell as to exert its effects [106,107,108,109]. In addition, researchers have made efforts to assess the influence of hydrophobicity on anticancer effects. For instance, Huang et al. [108] systematically altered the hydrophobicity of a 26-residue α-helical peptide (peptide **P**; KWKSFLKTFKSAKKTVLHTALKAISS) by replacing Ala residues with the more hydrophobic Leu residues as to increase its hydrophobicity or by changing Leu residues to Ala residues as to decrease its hydrophobicity. On the basis of these results, the authors observed a correlation between higher hydrophobic variants whereby the Leu-substituted peptides (A12L and A20L) exhibited greater anticancer activity against cancer cells with higher IC_50_ values than peptide P. Thus, peptides with higher hydrophobicity are assumed to penetrate deeper into the hydrophobic core of the cell membrane, thereby causing pores or channels on the cancer cell membrane and exhibiting greater anticancer activity. From these results, it can be inferred that the presence of hydrophobic residues on the α-helical structure of ACPs can enhance its anticancer properties.

#### 3.4.2. Peptides Forming an Amphipathic α-Helix Contributes to Anticancer Activity

Interestingly, four of the ten top-ranked informative PCPs consisting of ARGP820102, BEGF750101, ISOY800106 and CHOP780201 described helical properties of peptides such as helical potential [104], inner helix [110], frequency of helix end [111] and frequency of α-helix [112], respectively, as shown in Table 6. This supports the notion that the helical structure plays a principal role in governing the anticancer property of peptides. In addition, the α-helical conformation as adopted by many ACPs in biological membranes, is now regarded as a key determinant of anticancer activity [113]. BMAP-27 (GRFKRFRKKFKKLFKKLSPVIPLLHL) and BMAP-28 (GGLRSLGRKILRAWKKY GPIIVPIIRI) are well-known peptides derived from bovine sources having a cationic NH_2_ terminal that forms an amphipathic α-helix. They exert their cytotoxic activity against human leukemia cells by enhancing membrane permeability, which allows for an influx of Ca++ ions, thereby leading to apoptosis via DNA fragmentation [114]. In addition, the structures of Cecropin A (KWKLFKKIEKVGQNIRDGIIKAGPAVAVVGQATQIAK) and Cecropin B (KWKVFKKIEKMGRNIRNGIVKAGPAIAVLGEAKAL), which were first derived from insects, primarily consisted of two α-helix [115,116] while its amphipathic N-terminal (i.e., that is capable of interacting with anionic membrane components) is responsible for mediating the cytotoxic activity against cancer cells. On the other hand, the C-terminal is hydrophobic and it is postulated to facilitate peptide entry into the membrane thereby enabling oligomerization to occur, which subsequently leads to leakage of cell components due to pore formation and eventual cell death [117]. In addition, Srisailam et al. [118] designed a custom lytic peptide by modifying the cecropin B (CB) to contain two identical hydrophobic segments on both the N and the C terminals, thus disrupting its normal α-helical structure. The resulting peptide CB-3, was unable to effectively lyse cell membranes on cancer cells as well as bacterial cells as compared to the control peptide, therefore highlighting the significance of the peptide structure on its function. Interestingly, Moore et al. [119] uncovered that CB exhibited anticancer activities when tested against colon adenocarcinoma cells in vivo and cytotoxicity against multidrug-resistant breast and ovarian cancer cell lines in vitro.

Similarly, Magainins isolated from the skin of *Xenopus laevis* is comprised of a α-helical secondary structure with separate cationic and hydrophobic faces. Magainin-2 (GIGKFLHSAKKFGKAFVGEIMNS) causes lysis of both hematopoietic and solid tumor cell lines as while observing selective cytotoxic activity against several human bladder cell lines [120]. In addition, it was perceived using florescence spectrometry that magainins could lyse tumor cells by forming ion-conducting α-helical channels in the cancer cell membrane [121]. Melittin (GIGAVLKVLTTGLPALISWIKRKRQQ) is another well-known ACP that is derived from the venom of European honeybee (*Apis mellifera*) [122]. The N-terminal region of melittin is largely hydrophobic whereas the C-terminal contains positively-charged amino acid residues. Melittin form channels in lipid bilayers and is lytic against both cancer cells and normal cells [123]. Self-association of amphipathic α-helical monomers of melittin is largely suggested to be responsible for perturbations of membrane integrity thereby leading to cellular lysis via the barrel stave mechanism [124]. Furthermore, the only human cathelicidin-derived cationic ACP to date, LL-37 (LLGDFFRKSKEKIGKEFKRIVQRIKDFLRNLVPRTES) is present throughout the body and assumes a α-helical conformation that is comprised of both cationic and hydrophobic faces oriented in a parallel manner with lipid membranes. As such these peptides are suggested to engage in a carpet-like mechanism for its cytotoxicity [125]. However, unlike other ACPs, LL-37 is toxic to eukaryotic cells at slightly higher concentrations and thus lacks the selectivity of an ACP. Nevertheless, Okumura et al. [126] showed that a C-terminal fragment of LL-37 (hCAP18) was able to induce mitochondrial depolarization and apoptosis in human oral squamous carcinoma cells but not in healthy human gingival fibroblasts and human keratinocyte cells. In summary, it is evident that the amphipathicity associated with certain α-helical structures is pivotal for the elicitation of peptide anticancer activities.

#### 3.4.3. Peptides Forming β-Sheet Are Vital to Peptide Structure and Contribute to Anticancer Activity

As seen in Table 6, 3 out of the 20 top-ranked informative PCPs pertain to β-sheet structure (i.e., BEGF750102, BEGF750103 and CHOP78010) which correspond to properties such as conformational parameter of β-structure, conformational parameter of β-turn and normalized frequency of β-turn [110], respectively. Although, most of the well-known APCs are α-helical structures, some β-sheet ACPs have also been extensively studied. Many studies showed that defensins contain a group of closely related Cys-Arg rich cationic peptides. Out of these, human α- and β-defensins remain the most studied, comprising of six conserved Cys residues that form three intramolecular disulfide bridges between the N-terminal and the C-terminal regions of the peptide [127,128]. In the studies of [129,130], human neutrophil peptides (HNPs) -1 (ACYCRIPACIAGERRYGTCIYQGRLWAFCC), -2 (CYCRIPACIAGERRYGTCIYQGRLWAFCC) and -3 (DCYCRIPACIAGERRYGTCIY QGRLWAFCC) were shown to possess cytotoxic effects against several human and mouse tumor cells such as human B-lymphoma cells, human oral squamous carcinoma cells, and mouse teratocarcinoma cells. HNPs were also shown to kill cancer cells through membrane binding mediated by electrostatic interactions, followed by rapid collapse of the membrane potential and loss of membrane integrity [130,131]. In addition, Mader et al. [100,132] reported that bovine lactoferricin (LfcinB; FKCRRWQWRMKKLGAPSITCVRRAF) isolated from cow’s milk, represents a cationic ACP with amphipathic β-sheet configuration displaying anticancer activity against leukemia cells and various other carcinomas [100,132]. Moreover, LfcinB not only exerts its action by inducing apoptosis through direct disruption of the mitochondrial membrane, but is also capable of lysing the membrane itself, depending on the cancer cell type [133,134].

Furthermore, isolated from the hemocytes of horseshoe crab (*Tachypleus tridentatus*), the ACP tachyplesin I (KWCFRVCYRGICYRRCR) is arranged in two anti-parallel β-sheets that are held in place by two disulfide bonds [135]. Chen et al. [136] discovered that tachyplesin I killed cancer cells in a unique way by binding to hyaluronan on human prostate carcinoma cells, as well as to the C1q complement component in human serum, leading to the activation of the classical immune complement pathway. As a result, complement-mediated lysis of tachyplesin I-coated cancer cells was achieved. In a separate study, Adamia et al. [137] observed that the binding activity of tachyplesin I to C1q was dependent on its secondary structure which consists of two disulfide bonds formed by Cys residues at positions 3–16 and 7–12. Previously it has also been demonstrated that, the deletion of Cys residues from tachyplesin I altered its β-sheet structure to that of a linear form thus, disrupting its activity [138]. Therefore, from the aforementioned studies, it can be postulated that β-sheet is vital for maintaining the stability of peptide structures. As noticed in Table 5, our result of informative amino acids is consistent with the above-mentioned studies, where Cys is the second most important amino acid having a MDGI value of 19.71.

Nevertheless, further research is necessary to determine the key amino acids that could transform non-ACPs into ACPs and thus, allow the generation of a library of potentially novel ACPs. The design of effective ACPs from non-ACPs could possibly be based on the improvement of physicochemical properties, like hydrophobicity, positive charge, replacing the amino acid used or adding a functional motif with better activity towards the targeted cancer cells. In that regard, Tada et al. [139] revealed that the replacement of amino acid His to Arg in epidermal growth factor receptor (EGFR) binding peptide increased its anti-tumor activity via stronger binding affinity to the EGFR on cancer cells. Furthermore, the utilization of a motif containing the amino acids Trp, Met and Trp (WMW) was able to sensitize apoptosis and reduce the migratory effect on cancer cells which in turn could synergize the anti-cancer activity of peptides [140]. Therefore, it is of paramount importance to explore this area further.

### 3.5. Interpretable Rules Acquisition Obtained from the RF Model

Interpretable rules crucial for discriminating ACPs from non-ACPs were derived from the RF model built using AAC features on the benchmark dataset known as IR-ACP. The main advantages of these constructed rules are two folds: (i) to demonstrate which amino acid or which combination of amino acids are effective for ACP prediction, and (ii) to simply discriminate ACPs from non-ACPs without the need to go through the mathematical and computational details. Table 7 and Table A2 shows the twenty interpretable rules that were important for ACPs and eleven that were important for non-ACPs.

Table 7 presents eight interpretable rules that were important for identifying ACPs and each rule comprises a different number of criteria. If a query peptide meets all of the criteria in at least one of the eight rules, then it will be predicted as an ACP. As observed in Table 7, four rules afforded a prediction accuracy larger than 95%, namely rules #1, #2, #3 and #8, while the remaining rules provided acceptable results with accuracy ranging between 80.79% and 87.88%. These results indicated that the constructed rules are robust, both in terms of their accuracy and interpretability for predicting and characterizing ACPs.

### 3.6. ACPred Web Server

In order to make the prediction model presented herein a practical tool that can be widely used by the scientific community, we have constructed a web server called the ACPred using the best model as described in previous sections. The web interface has been established using the Shiny package under the R programming environment. The web server is freely accessible at http://codes.bio/acpred/. Screenshots of the ACPred web server is shown in Figure 5 in which panel A shows the web server prior to submission of input data and panel B shown the web server after the prediction has been made.

Briefly, a step-by-step guide on using the web server is given below:Step 1. Enter the following URL into the web browser, http://codes.bio/acpred/.Step 2. Either enter the query peptide sequence into the Input box or upload the sequence file by clicking on the “Choose file” button (i.e., found below the “Enter your input sequence(s) in FASTA format heading”).Step 3. Click on the “Submit” button to initiate the prediction process.Step 4. Prediction results are automatically displayed in the grey box found below the “Status/Output” heading. Typically, it takes a few seconds for the server to process the task. Users can also download the prediction results as a CSV file by pressing on the “Download CSV button”.

Additionally, users can also run a local copy of ACPred on their own computer by using the following one-line code in an R environment:

shiny::runGitHub(‘acpred-webserver’, ‘chaninlab’)

However, prior to running the aforementioned code, it is recommended that users first install the prerequisite R packages. This can be performed by using the following code:

install.packages(c(‘shiny’, ‘shinyjs’, ‘shinythemes’, ‘protr’, ‘seqinr’, ‘caret’, ‘markdown’))

## 4. Conclusions

There is increasing evidence indicating that ACPs could be a potential, valuable therapeutic agent for cancer treatment. With the avalanche of peptide sequences generated from many sources, there is significant demand for computational methods that can support the timely discrimination of ACPs from other peptides, as well as offer insights from analysis using only their primary sequences. In this study, we have developed a the ACPred model in a systematic manner by taking advantage of two powerful machine learning methods consisting of support vector machine (SVM) and random forest (RF) for the prediction and characterization of anticancer activities of peptides. The best model for further development of ACPred was achieved by using the SVM method coupled with the combination of amino acid composition and pseudo amino acid composition features. By comparing with the state-of-the-art method, the TargetACP, it was found that the performance of ACPred was comparable to that of TargetACP (95.61% vs. 96.22% accuracies) as assessed by the rigorous jackknife test. Moreover, the feature importance analysis and interpretable rules extraction were carried out by using the RF model to identify which feature was the most important in predicting ACPs while also providing a better understanding of the anticancer activities of investigated peptides. Results from model interpretation revealed that the hydrophobic residues on the α-helical structure, Cys residues on the β-sheet structure and the formation of amphipathic α-helix played a crucial role in the anticancer activity of the investigated peptides. Finally, a web server named ACPred was established and made freely available online at https://codes.bio/acpred/.

## Figures and Tables

**Figure 1 molecules-24-01973-f001:**
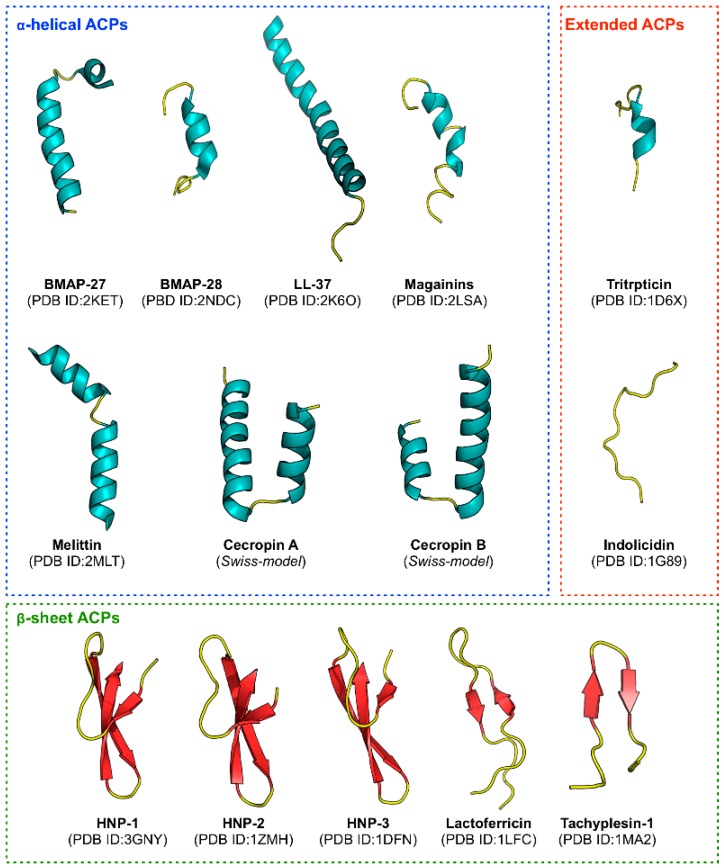
Overview of the structural diversity of three classes of anticancer peptides. Each structure is labeled by its common name followed by the Protein Data Bank (PDB ID) in parenthesis on the subsequent line. In cases where the PDB ID was not available, the SWISS-MODEL server (available at: https://swissmodel.expasy.org/) was used to construct the structure.

**Figure 2 molecules-24-01973-f002:**
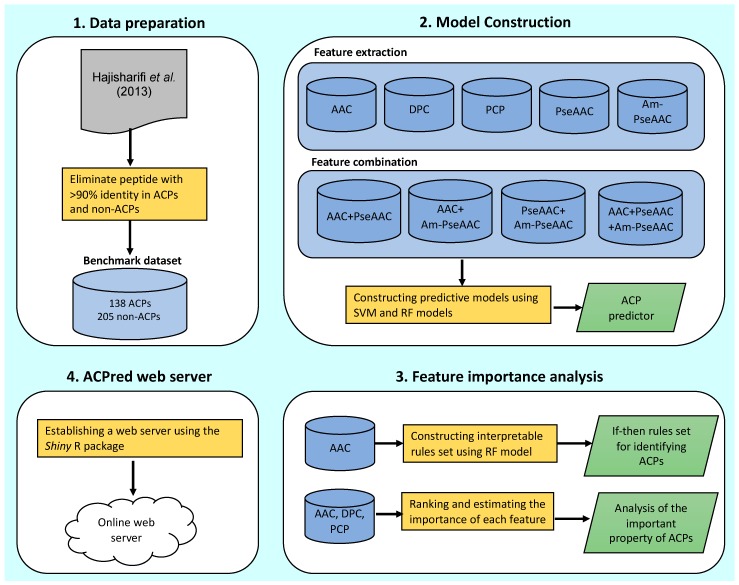
Schematic framework of ACPred.

**Figure 3 molecules-24-01973-f003:**
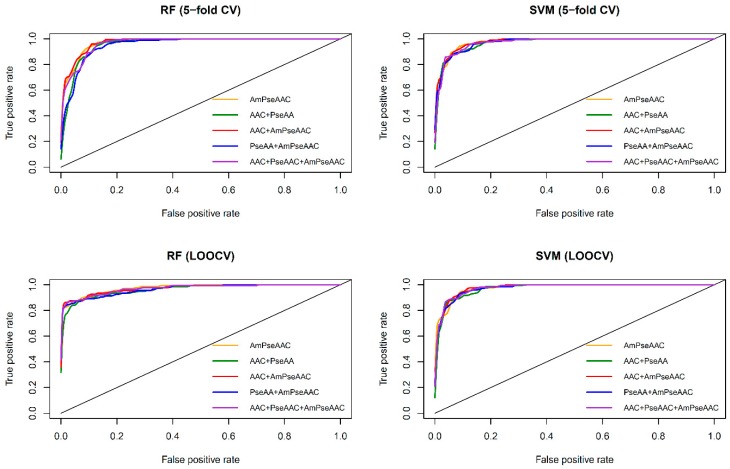
ROC curve of RF (top and bottom left) and SVM (top and bottom right) models as assessed by 5-fold cross-validation (top left and right) and jackknife test or leave-one-out cross-validation (bottom left and right).

**Figure 4 molecules-24-01973-f004:**
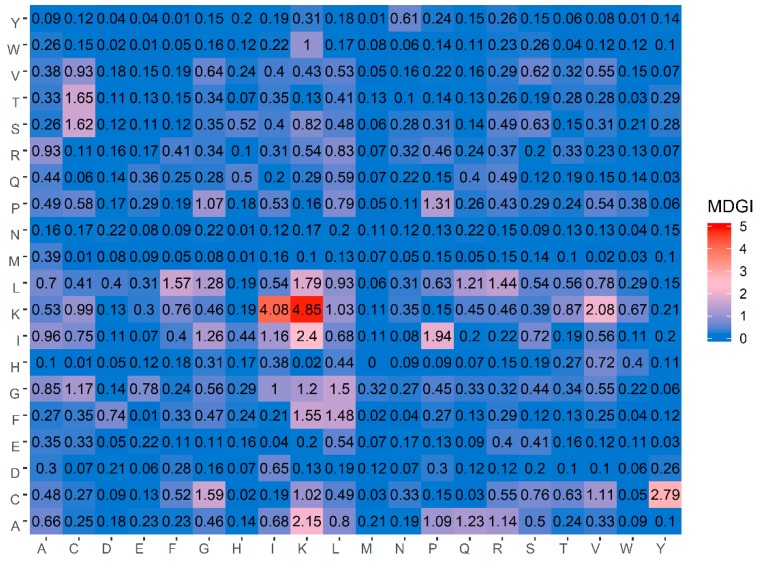
Heat map of the mean decrease of Gini index of dipeptide compositions.

**Figure 5 molecules-24-01973-f005:**
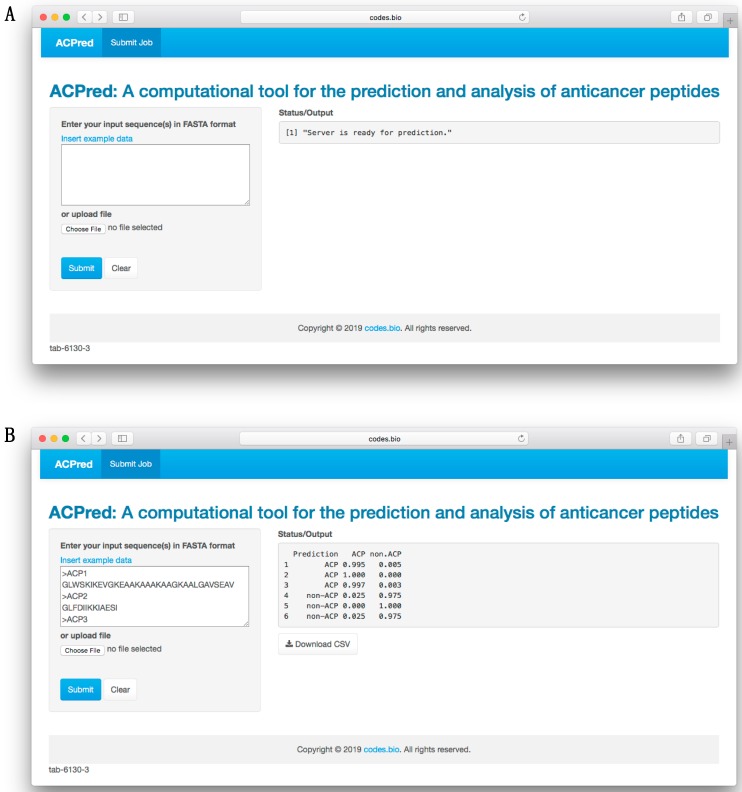
Screenshots of the ACPred web server before (**A**) and after (**B**) submission of sequence data for prediction.

**Table 1 molecules-24-01973-t001:** Summary of existing methods for predicting anticancer peptides.

Method (Year)	Classifier ^a^	Sequence Features ^b^	Interpretable	Web Server
AntiCP (2013)	SVM	AAC, DPC, BP	No	✓
Hajisharifi et al. (2014)	SVM	PseACC, LAK	No	
ACPP (2015)	SVM	PRM	No	✓ ^c^
iACP (2016)	SVM	g-gap DPC	**Yes**	✓
Li and Wang (2016)	SVM	AAC, RACC, acACS	No	
iACP-GAEnsC (2017)	Ensemble method	Pse-g-gap DPC, Am-PseAAC, RACC	No	
MLACP (2017)	RF	AAC, ATC, DPC, PCP	**Yes**	✓
SAP (2018)	SVM	g-gap DPC	No	
TargetACP (2018)	SVM	CPSR, SAAC, PsePSSM	No	
ACPred (this study)	SVM	AAC, DPC, PCP, PseAAC, Am-PseAAC	**Yes**	✓

^a^ RF: random forest, SVM: support vector machine. ^b^ AAC: amino acid composition, ATC: atomic composition, acACS: auto covariance of the average chemical shift, Am-PseAAC: amphiphilic pseudo amino acid composition, BP: binary profile, CPDR: composite protein sequence representation, DPC: dipeptide composition, g-gap DPC: G-Gap dipeptide composition, LAK: local alignment kernel, PCP: Physicochemical properties, PseACC: Pseudo amino acid composition, Pse-g-gap DPC: Pseudo G-Gap dipeptide composition, PRM: protein relatedness measure, RACC: reduce amino acid composition, SAAC: split amino acid composition. ^c^ The web server is not accessible.

**Table 2 molecules-24-01973-t002:** Performance comparison of SVM and RF with various types of sequence features over five-fold cross-validation.

Feature	Classifier	*Ac (%)*	*Sn (%)*	*Sp (%)*	*MCC*	*auROC*
AAC	SVM	92.69	83.94	98.54	0.850	0.977
	RF	91.23	92.80	90.32	0.817	0.958
DPC	SVM	83.92	100.00	78.85	0.687	0.942
	RF	87.14	91.89	84.85	0.733	0.944
PCP	SVM	84.80	63.50	99.02	0.698	0.938
	RF	83.63	90.10	80.91	0.661	0.872
PseAAC	SVM	92.98	84.67	98.54	0.856	0.990
	RF	92.11	93.65	91.20	0.835	0.959
Am-PseAAC	SVM	95.03	87.59	100.00	0.899	0.995
	RF	92.40	100.00	88.75	0.848	0.974
AAC + PseAAC	SVM	93.57	85.40	99.02	0.869	0.991
	RF	92.98	96.69	90.95	0.855	0.964
AAC + Am-PseAAC	SVM	95.32	89.05	99.51	0.904	0.994
	RF	93.28	97.50	90.99	0.862	0.969
PseAAC + Am-PseAAC	SVM	94.44	86.86	99.51	0.887	0.994
	RF	92.69	98.28	89.82	0.851	0.967
AAC + PseAAC + Am-PseAAC	SVM	94.15	86.13	99.51	0.881	0.993
	RF	92.98	98.29	90.22	0.857	0.972

**Table 3 molecules-24-01973-t003:** Performance comparison of SVM and RF with various types of sequence features over jackknife test.

Feature	Classifier	*Ac (%)*	*Sn (%)*	*Sp (%)*	*MCC*	*auROC*
AAC	SVM	92.98	83.94	99.02	0.857	0.978
	RF	91.23	92.80	90.32	0.817	0.959
DPC	SVM	85.09	98.86	80.32	0.706	0.941
	RF	86.84	89.66	85.40	0.725	0.947
PCP	SVM	84.80	63.50	99.02	0.698	0.937
	RF	83.63	88.57	81.44	0.659	0.868
PseAAC	SVM	92.98	84.67	98.54	0.856	0.990
	RF	93.28	97.50	90.99	0.862	0.960
Am-PseAAC	SVM	95.03	87.59	100.00	0.899	0.995
	RF	92.40	99.12	89.08	0.847	0.969
AAC + PseAAC	SVM	93.57	85.40	99.02	0.869	0.990
	RF	93.28	98.31	90.63	0.863	0.962
AAC + Am-PseAAC	SVM	95.61	89.78	99.51	0.910	0.994
	RF	93.57	98.32	91.03	0.869	0.967
PseAAC + Am-PseAAC	SVM	93.86	85.40	99.51	0.875	0.992
	RF	93.57	99.15	90.67	0.870	0.959
AAC + PseAAC + Am-PseAAC	SVM	94.74	87.59	99.51	0.893	0.994
	RF	92.98	99.13	89.87	0.858	0.973

**Table 4 molecules-24-01973-t004:** Performance comparison of the proposed ACPred model with existing methods.

Method ^a^	*Ac (%)*	*Sn (%)*	*Sp (%)*	*MCC*
Hajisharifi et al. ^b^	92.68	89.70	85.18	0.78
iACP ^b^	95.06	89.86	98.54	0.90
iACP-GAEnsC ^b^	96.45	95.36	97.54	0.91
TargetACP ^b^	96.22	94.20	97.57	0.92
TargetACP ^c^	98.78	99.02	98.54	0.97
ACPred ^b^	95.61	89.78	99.51	0.91
ACPred-modified ^c^	97.56	96.08	99.02	0.95

^a^ Results were reported from the work of TargetACP. ^b^ Results were performed on the benchmark dataset consisting of 138 ACPs and 205 non-ACPs. ^c^ Results were performed on the balanced dataset consisting of 205 ACPs and 205 non-ACPs by using the SMOTE technique on the benchmark dataset.

**Table 5 molecules-24-01973-t005:** Amino acid compositions (%) of anticancer and non-anticancer peptides along with their difference as well as MDGI values. The rank of each amino acid amongst their 20 amino acids are shown in parenthesis for AAC difference and MDGI.

Amino Acid	ACP (%)	Non-ACP (%)	Difference	*p*-Value	MDGI
A-Ala	7.623	11.005	−3.383 (7)	<0.05	6.41 (11)
C-Cys	3.906	8.015	−4.109 (5)	<0.05	19.71 (2)
D-Asp	2.417	3.418	−1.002 (15)	<0.05	3.85 (15)
E-Glu	1.707	3.523	−1.816 (12)	<0.05	6.88 (9)
F-Phe	6.823	2.41	4.413 (2)	<0.05	6.12 (12)
G-Gly	1.975	4.123	−2.148 (11)	<0.05	6.56 (10)
H-His	1.536	5.798	−4.262 (4)	<0.05	7.80 (7)
I-Ile	10.072	6.98	3.092 (9)	<0.05	10.13 (5)
K-Lys	2.542	1.651	0.892 (17)	0.086	29.54 (1)
L-Leu	8.099	3.739	4.36 (3)	<0.05	7.84 (6)
M-Met	9.831	13.888	−4.057 (6)	<0.05	4.34 (14)
N-Asn	11.497	3.964	7.533 (1)	<0.05	4.98 (13)
P-Pro	0.905	2.224	−1.319 (13)	<0.05	7.37 (8)
Q-Gln	5.385	2.711	2.674 (10)	<0.05	15.89 (3)
R-Arg	4.211	7.495	−3.283 (8)	<0.05	11.42 (4)
S-Ser	6.537	5.832	0.705 (19)	0.245	3.50 (16)
T-Thr	3.781	4.704	−0.923 (16)	0.098	3.13 (17)
V-Val	2.258	1.560	0.698 (20)	0.083	2.96 (18)
W-Trp	2.244	1.423	0.821 (18)	<0.05	2.51 (20)
Y-Tyr	6.65	5.539	1.111 (14)	0.091	2.90 (19)

**Table 6 molecules-24-01973-t006:** Ten top-ranked physiocochemical properties from the AAindex having the highest MDGI values.

Rank	AAindex	Categorized Property	Description	MDGI
1	ARGP820101	Hydrophobicity	Hydrophobicity index (Argos et al., 1982)	1.51
2	ARGP820102	Hydrophobicity	Signal sequence helical potential (Argos et al., 1982)	1.40
3	BHAR880101	Hydrophobicity	Average flexibility indices (Bhaskaran-Ponnuswamy, 1988)	1.08
4	ARGP820103	Hydrophobicity	Membrane-buried preference parameters (Argos et al., 1982)	1.04
5	BEGF750102	Beta propensity	Conformational parameter of beta-structure (Beghin-Dirkx, 1975)	1.00
6	BEGF750101	Alpha and turn propensities	Conformational parameter of inner helix (Beghin-Dirkx, 1975)	0.92
7	BIGC670101	Physicochemical properties	Residue volume (Bigelow, 1967)	0.91
8	BEGF750103	Alpha and turn propensities	Conformational parameter of beta-turn (Beghin-Dirkx, 1975)	0.87
9	BIOV880102	Hydrophobicity	Information value for accessibility; average fraction 23% (Biou et al., 1988)	0.85
10	ISOY800107	Hydrophobicity	Normalized relative frequency of double bend (Isogai et al., 1980)	0.83

**Table 7 molecules-24-01973-t007:** Eight if-then rules for the prediction of anticancer peptides using random forest and amino acid composition.

No.	Rule	Covered Samples	Misclassified Sample	Ac (%)
1	G > 0.041 and I > 0.0615 and L ≤ 0.1715 and K > 0.0385 and M ≤ 0.027	48	1	97.92
2	R ≤ 0.0515 and Q ≤ 0.026 and K > 0.094	48	1	97.92
3	C > 0.1145 and P ≤ 0.073	35	0	100.00
4	L ≤ 0.093 and F > 0.0715 and S ≤ 0.152	33	4	87.88
5	A ≤ 0.0145 and Q ≤ 0.026	26	4	84.62
6	R ≤ 0.0665 and E ≤ 0.044 and H > 0.052	21	4	80.95
7	I > 0.108 and K > 0.055	42	6	85.71
8	E ≤ 0.0545 and G > 0.0365 and K > 0.108 and M ≤ 0.04	49	1	97.96

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
