# Peer review of "ACPred: A Computational Tool for the Prediction and Analysis of Anticancer Peptides"

_molecules, 2019, doi:10.3390/molecules24101973_

Round 1
Reviewer 1 Report
The authors present a comprehensive write-up of the development and implementation of a classification algorithm for anti-cancer peptides (ACPs) – denoted ACPred. ACPs are short amino-acid sequence proteins that have therapeutic potential for treating cancer. The authors propose the use of a variety of AA sequence-based features and physico-chemical properties. A variety of state-of-the-art machine learning algorithms were employed along with varying combinations of features.
Overall, the manuscript is very well-organized, well-written, and comprehensive in its content. The authors should also be commended for their dedication to transparency and reproducibility, including the public availability of the complete code and benchmarking datasets. These are particularly valuable in machine learning/predictive modeling, where the dangers of over-fitting and inflating performance claims are quite real. The authors also supply a web-based software tool to maximize the utility of their algorithm. However, I do have some questions/comments that warrant consideration, particularly on model training/selection vis a vis the performance assessment.
Major Comments:
Some clarity on the model tuning may be useful, specifically if the 5-fold CV was used within the training set for tuning parameter selection. In other words, for the 5-fold CV approach to performance evaluation, within any given set of 4 folds used for training – this was further divided into 5 sub-folds for selecting the tuning parameters of the respective algorithms (i.e., nested cross-validation). If this isn’t the case, and the same CV process is used for both tuning AND evaluation, this can result in biased performance estimates (see Varma and Simon 2006).
In a similar vein, a large amount of models are considered (Table 2), and an optimal one is selected among these. It seems this model selection is based on the same performance metrics used in the methods comparison in Table 4. This would seem to warrant a separate independent validation set to provide honest method comparisons. Useful discussion in this regard can be found in arxiv paper by Raschka (https://arxiv.org/pdf/1811.12808.pdf).
Minor Comments:
In Figure 3, the bottom left figure (RF LOOCV) labels both the x and y axes as “True Positive Rate” Presumably the x-axis is in error (actually FPR).
On page 7, lines 237-239, the authors state the SVM models can deal with over-fitting through the use of kernels (“mapping the input samples to a higher dimension space and then searching for the maximum-margin hyperplane”). While kernels provide model flexibility through the exploration of the higher-order feature space, I would not say that helps with overfitting in anyway (in fact, quite the opposite).
It wasn’t clear to me what the 3 benchmark datasets were from the description in the Reproducibility section of the text. What is there above and beyond the one discussed in the text?
Author Response
Point 1: Overall, the manuscript is very well-organized, well-written, and comprehensive in its content. The authors should also be commended for their dedication to transparency and reproducibility, including the public availability of the complete code and benchmarking datasets. These are particularly valuable in machine learning/predictive modeling, where the dangers of over-fitting and inflating performance claims are quite real. The authors also supply a web-based software tool to maximize the utility of their algorithm. However, I do have some questions/comments that warrant consideration, particularly on model training/selection vis a vis the performance assessment.
Response 1: Thank you for the encouraging and kind words of inspiration
Point 2: Some clarity on the model tuning may be useful, specifically if the 5-fold CV was used within the training set for tuning parameter selection. In other words, for the 5-fold CV approach to performance evaluation, within any given set of 4 folds used for training – this was further divided into 5 sub-folds for selecting the tuning parameters of the respective algorithms (i.e., nested cross-validation). If this isn’t the case, and the same CV process is used for both tuning AND evaluation, this can result in biased performance estimates (see Varma and Simon 2006).
Response 2: Thank you for the kind suggestion of this point. From the suggested article, they reported that a nested CV procedure provides more unbiased estimate of the true error as compared to the conventional CV procedure (5- or 10-fold CV). Although, the nested CV procedure is important for improving the validity and robustness of our proposed model, however to perform a fair comparison with the existing methods (as shown in Table 1), our proposed model will have to use the same validation method as that of previous methods thus supporting the use of 5-fold CV and LOOCV in this study.
Point 3: In a similar vein, a large amount of models are considered (Table 2), and an optimal one is selected among these. It seems this model selection is based on the same performance metrics used in the methods comparison in Table 4. This would seem to warrant a separate independent validation set to provide honest method comparisons. Useful discussion in this regard can be found in arxiv paper by Raschka (https://arxiv.org/pdf/1811.12808.pdf).
Response 3: Thank you for the kind suggestion of this point. In fact, the independent dataset test can be treated as a special case of the sub-sampling test [41]. As described in a series of work from the group of KC Chou [41, 73, 74], from amongst the three cross-validation methods, the jackknife test is considered to be one of the most rigorous methods for cross-validation in statistics and can provide a unique result for a given benchmark dataset. In fact, for the jackknife cross-validation, each samples in the dataset is left out as an independent test sample and then all parameters are calculated based on the remaining samples without including the one being identified.
Point 4: In Figure 3, the bottom left figure (RF LOOCV) labels both the x and y axes as “True Positive Rate” Presumably the x-axis is in error (actually FPR).
Response 4: Thank you for the kind suggestion. This point has been corrected as suggested.
Point 5: On page 7, lines 237-239, the authors state the SVM models can deal with over-fitting through the use of kernels (“mapping the input samples to a higher dimension space and then searching for the maximum-margin hyperplane”). While kernels provide model flexibility through the exploration of the higher-order feature space, I would not say that helps with overfitting in anyway (in fact, quite the opposite).
· Response 5: Thank you for the kind suggestion for this point. As mentioned in the study of Tung et al. and Lameski et al., they investigated the impact of parameter tuning of SVMs via grid search approach for preventing over-fitting. The results showed that SVM with proper parameter tuning can not only improve the predictive performances, but also increase stability of the classification models on both training and testing dataset. It could be stated that SVMs with proper parameter tuning could cope well the over-fitting problem.
References
Tung, C. W., Ziehm, M., Kämper, A., Kohlbacher, O., & Ho, S. Y. (2011). POPISK: T-cell reactivity prediction using support vector machines and string kernels. BMC bioinformatics, 12(1), 446.
Lameski, P., Zdravevski, E., Mingov, R., & Kulakov, A. (2015). SVM parameter tuning with grid search and its impact on reduction of model over-fitting. In Rough Sets, Fuzzy Sets, Data Mining, and Granular Computing (pp. 464-474). Springer, Cham.
Point 6: It wasn’t clear to me what the 3 benchmark datasets were from the description in the Reproducibility section of the text. What is there above and beyond the one discussed in the text?
Response 6: Thank you for the kind suggestion. After checking, we found that there is indeed only one benchmark dataset used in this study. Therefore, we have corrected the number from three to one in the revised version as follows:
all R codes and the three benchmark datasets…
has been changed to
all R codes and the benchmark dataset.
Reviewer 2 Report
The authors have used a machine learning-based approach to classify fasta sequence inputs as either ACP (anti-cancer peptide) probability score and a (1-probability) of it not being ACP.
The paper is very tidy and concisely written with a nice flow of presentation and definitely deserves publication but I have a few comments/ concerns.
Major:
1) The authors demonstrate the inclusion of 5 feature parameters in Table 1 and show that- in comparison to other methods, this has most number of features thus adding robustness to the algorithm. However, I was expecting a discussion on what could be a possible improvement on the current method, what could be new features which when included would help eliminate false positives. Any comments on that would be useful.
2) In the beginning the authors mention that positively charged amino acids in these ACPs are key to neutralizing the anionic surface of cancer cells. However, Table 5 shows that arginines and lysines comprise 4.2% and 2.5% of ACPs respectively. This is lesser than polar groups like serine which are seen to comprise ~6% of ACPs. This makes me think if it is actually the positively charged groups or is it the potential form a hydrogen bond which is sufficient to neutralize the cancer cells. Any discussion on this is key to substantiate claims made in the beginning of the paper. I further think it is absolutely important to alleviate any equivocation in the manuscript to deem it publishable.
3) Finally, this comment is more of a suggestion.
The authors could include a short analysis to identify one or two mutations in non-ACPs that would make them ACPs. This would be more like an affinity maturation and thus researchers working with various peptides can have an idea if its possible to generate a library of potential ACPs starting with a non-ACP.
This would be akin to a paper I reviewed for MDPI where the authors make an automated tool for rapid de novo antibody generation against any antigen structure (OptMaven: https://www.mdpi.com/2073-4468/7/3/23). The authors use mixed integer linear optimization to further suggest single amino acid changes to any antibody sequence to make it more humanized (to eliminate immunogenic interactions) - which is deployable as a standalone module.
Minor:
4) I think the server output of ACP fraction (x) is sufficient as the non-ACP fraction is just (1-x).
Author Response
Point 1: The paper is very tidy and concisely written with a nice flow of presentation and definitely deserves publication but I have a few comments/ concerns.
Response 1: Thank you for the encouraging and kind words of inspiration.
Point 2: The authors demonstrate the inclusion of 5 feature parameters in Table 1 and show that- in comparison to other methods, this has most number of features thus adding robustness to the algorithm. However, I was expecting a discussion on
what could be a possible improvement on the current method,
what could be new features which when included would help eliminate false positives.
Any comments on that would be useful.
Response 2: Thank you for the kind suggestion of this point.
As suggested, we have added possible approaches for improvement of the existing methods for predicting anticancer peptides as our current approach already demonstrates improvements upon existing ones.
As suggested, we have added an in-depth discussion rationalizing the need for new peptide features while also reviewing available peptide features and their utility in the field of peptide QSAR.
Point 3: In the beginning the authors mention that positively charged amino acids in these ACPs are key to neutralizing the anionic surface of cancer cells. However, Table 5 shows that arginines and lysines comprise 4.2% and 2.5% of ACPs respectively. This is lesser than polar groups like serine which are seen to comprise ~6% of ACPs. This makes me think if it is actually the positively charged groups or is it the potential form a hydrogen bond which is sufficient to neutralize the cancer cells. Any discussion on this is key to substantiate claims made in the beginning of the paper. I further think it is absolutely important to alleviate any equivocation in the manuscript to deem it publishable.
Response 3: Thank you for your suggestion. Since the intrinsic property of ACPs include its cationicity, we chose to focus our results and discussion on this specific property while also highlighting other properties of ACPs using their physicochemical properties. The suggested point has been added to the Results and Discussion part from line 460-473.
Point 4: The authors could include a short analysis to identify one or two mutations in non-ACPs that would make them ACPs. This would be more like an affinity maturation and thus researchers working with various peptides can have an idea if its possible to generate a library of potential ACPs starting with a non-ACP.
This would be akin to a paper I reviewed for MDPI where the authors make an automated tool for rapid de novo antibody generation against any antigen structure (OptMaven: https://www.mdpi.com/2073-4468/7/3/23). The authors use mixed integer linear optimization to further suggest single amino acid changes to any antibody sequence to make it more humanized (to eliminate immunogenic interactions) - which is deployable as a standalone module.
Response 4: Thank you for providing us with a good idea as per the article suggested. It is indeed possible to apply this model for the design of effective ACPs from non-ACPs based on the improvement of physicochemical properties, like hydrophobicity, positive charge, replacing the amino acid used or adding a functional motif with better activity towards the targeted cancer cells. However, the construction of an automated tool that could potentially change non-ACPs to ACPs via mutation of amino acids would be an extended effort that we look forward to exploring in the future. In the current research, the change in amino acid for better functioning of ACPs has already been discussed in the paper under the subheading 3.4.1. In addition, we have added a few more examples in the article at line 685-695.
Point 5: I think the server output of ACP fraction (x) is sufficient as the non-ACP fraction is just (1-x).
Response 5: Yes, this is true that the result for non-ACP is 1-x given x is the probability of a sequence being an ACP. However, the major purpose of the server output is to return the prediction results for each row of input sequence whether they are ACP or non-ACP while the probability value that follows is an auxiliary parameter that informs the user of the confidence that they can place on each of the predicted sequence activity. Thus, in our humble opinion both output values consisting of (i) prediction of being ACP or non-ACP and (ii) probability value are essential for the users.

Round 2
Reviewer 2 Report
The authors have responded satisfactorily